# New Insights into the Mechanism of Immune-Mediated Tissue Injury in Yellow Fever: The Role of Immunopathological and Endothelial Alterations in the Human Lung Parenchyma

**DOI:** 10.3390/v14112379

**Published:** 2022-10-27

**Authors:** Danielle Barbosa Vasconcelos, Luiz Fabio Magno Falcão, Lucas Coutinho Tuma da Ponte, Camilla Costa Silva, Livia Caricio Martins, Bruno Tardelli Diniz Nunes, Arnaldo Jorge Martins Filho, Edna Cristina Santos Franco, Maria Irma Seixas Duarte, Jorge Rodrigues de Sousa, Pedro Fernando da Costa Vasconcelos, Juarez Antônio Simões Quaresma

**Affiliations:** 1Secção de Arbovirologia e Febres Hemorrágicas, Instituto Evandro Chagas (IEC), Ananindeua 67015-120, Brazil; 2Departamento de Patologia, Universidade do Estado do Pará (UEPA), Belém 66050-540, Brazil; 3Faculdade de Medicina, Universidade de São Paulo (USP), São Paulo 66075-110, Brazil; 4Núcleo de Medicina Tropical, Universidade Federal do Pará (UFPA), Belém 05508-070, Brazil

**Keywords:** yellow fever, lung, vascular endothelium, immune response

## Abstract

Yellow fever (YF) may cause lesions in different organs. There are no studies regarding the in situ immune response in the human lung and investigating immunopathological aspects in fatal cases can help to better understand the evolution of the infection. Lung tissue samples were collected from 10 fatal cases of human yellow fever and three flavivirus-negative controls who died of other causes and whose lung parenchymal architecture was preserved. In YFV-positive fatal cases, the main histopathological changes included the massive presence of diffuse alveolar inflammatory infiltrate, in addition to congestion and severe hemorrhage. The immunohistochemical analysis of tissues in the lung parenchyma showed significantly higher expression of E-selectin, P-selectin, ICAM-1, VCAM-1 in addition to cytokines such as IL-4, IL-10, IL-13, TNF- α, IFN-γ and TGF-β compared to the negative control. The increase in immunoglobulins ICAM-1 and VCAM-1 results in strengthening of tissue transmigration signaling. E-selectin and P-selectin actively participate in this process of cell migration and formation of the inflammatory infiltrate. IFN-γ and TNF-α participate in the process of cell injury and viral clearance. The cytokines IL-4 and TGF-β, acting in synergism, participate in the process of tissue regeneration and breakdown. The anti-inflammatory cytokines IL-4, IL-10 and IL-13 also act in the reduction of inflammation and tissue repair. Our study indicates that the activation of the endothelium aggravates the inflammatory response by inducing the expression of adhesion molecules and cytokines that contribute to the rolling, recruitment, migration and eliciting of the inflammatory process in the lung parenchyma, contributing to the fatal outcome of the disease.

## 1. Introduction

Yellow fever (YF) is an acute febrile infectious disease endemic to tropical regions of Africa and South America, caused by the yellow fever virus (YFV), which belongs to the *Flaviviridae* family and *Flavivirus* genus [1].

As it is a hemorrhagic disease with pan-systemic behavior, studies have shown that the evasion strategies adopted by the virus contribute to the imbalance of the immune response, especially with regard to the expression of pro and anti-inflammatory cytokines that can influence the clinical evolution of disease, exacerbate tissue damage and cause massive destruction of resident cells in various organs such as the liver, heart and kidneys [2,3,4,5,6,7,8].

The lung has a highly integrated vascular network and, based on findings from experimental infection in non-human primates, the local microenvironment triggers an immune response that causes tissue aggression and, as a result, we observe vascular changes that lead to hemorrhage and inflammation with tissue damage [6,7,8,9]. Moreover, suppurative bronchopneumonia with alveolar damage can be observed in the lung along with diffuse oedema that can be noticed macroscopically and microscopically [7]. Despite this characteristic picture, no studies regarding the mechanism of cell injury, tissue immune response and the relationship between these two phenomena to the contribution of lung lesions in YF are available. Therefore, we aimed to characterize the pulmonary histopathological changes associated with the host immune response pattern in fatal YFV infections in humans. The findings of our study provide new insights into YF pathogenesis, especially the immunopathogenesis of lung injury in fatal YFV infections, and can help identify putative therapeutic targets and develop effective vaccines with fewer adverse effects.

## 2. Methods

### 2.1. Patients, Samples, and Diagnostic

Lung tissue samples were selected from 10 fatal cases of human yellow fever available in the archives of the Department of Pathology of Instituto Evandro Chagas (Belém, Brazil). Confirmation of the diagnosis for positive cases of YF included blood, liver and spleen samples that underwent histopathological analysis, immunohistochemistry tests and transcription-polymerase chain reaction (RT-qPCR) following a protocol adapted from Domingo et al. [10] and Menting et al. [11]. Three cases were part of the control group, had preserved lung parenchyma architecture and were negative for the most common flaviviruses circulating in Brazil (DENV and ZIKV) and respiratory virus infections. More information on patient demographics can be found in Table 1.

### 2.2. Ethical Aspects

The experimental protocols followed in this study were approved by the Research Ethics Committee of the Instituto Evandro Chagas (number 3.544.874), Ananindeua, Pará, Brazil and were in accordance with the recommendations provided by the Conselho Nacional de Ética em Pesquisa of the Conselho Nacional de Saúde (number 466/2012).

### 2.3. Immunohistochemical Assay

For the detection of phenotypic markers and cytokines antibodies specific for E-selectin (1:100) (Novus Biologicals, Centennial, CO, USA, NBP1-40109), P-selectin (1:100) (Novus Biologicals, NBP2-22046), intracellular adhesion molecule-1 (ICAM-1) (1:100) (RD Systems, Minneapolis, MN, USA, BBA19), vascular cell adhesion molecule (VCAM-1) (1:100) (Abcam, Cambridge, UK, ab 134047), interleukin-4 (IL-4) (1:100) (Abcam/9622), interleukin-10 (IL-10) (1:100) (Abcam/34843), interleukin-13 (IL-13) (1:100) (Abcam/9576), tumor necrosis factor-α (TNF-α) (1:100) (Abcam/6671), interferon-γ (IFN-γ) (1:100) (Biorbyt/orb10878) and transforming growth factor-β (TGF-β) (1:100) (Abcam/190503), the immunohistochemical assay was based on biotin-streptavidin peroxidase complex formation. For immunolabeling, lung tissue samples embedded in paraffin were dewaxed in xylene and hydrated in 90, 80 and 70% ethyl alcohol. Endogenous peroxidase activity was blocked with 3% hydrogen peroxide for 45 min. Antigen retrieval was performed using citrate buffer (pH 6.0) for 20 min at 90 °C and non-specific proteins were blocked with 10% skim milk for 30 min. Primary antibodies were diluted in 1% bovine serum albumin for 14 h. The biotinylated secondary antibodies LSAB (DakoCytomation, Glostrup, Denmark) were then added and incubated for 30 min at 37 °C, followed by streptavidin peroxidase (DakoCytomation) for 30 min at 37 °C. Antibody binding was visualized using a chromogen solution composed of 0.03% diaminobenzidine and 3% hydrogen peroxide and the tissue sections were counterstained with Harris haematoxylin for 1 min. Finally, the sections were dehydrated in a series of ethanol solutions of increasing concentrations and clarified in xylene.

### 2.4. Quantitative Analysis and Photodocumentation

The tissue sample slides were analysed using an Axio Imager Z1 microscope (Zeiss, Oberkochen, Germany). Immunostaining was quantitatively evaluated by randomly selecting 10 fields at high magnification (400×) in the lung parenchyma. Each field was subdivided into cross-sections of 10 × 10 subdivisions delimited by a grid, comprising an area of 0.0625 mm^2^.

### 2.5. Statistical Analysis

Data were stored in a spreadsheet and analysed using GraphPad Prism software (GraphPad Software Inc., San Diego, CA, USA). Numerical variables were expressed as means and standard deviations. The hypotheses were tested using the Mann–Whitney test and Spearman correlation while considering a significance level of 5% (*p* ≤ 0.05).

## 3. Results

Lung tissue analysis of patients with fatal YF revealed characteristic morphological changes. The main histopathological findings were capillary leakage with intense hemorrhagic foci and an inflammatory infiltration by lymphocytes, plasma cells and rare neutrophils in the lung parenchyma, which either filled the lumen or were distributed along the wall of the alveoli (Figure 1).

P-selectin, E-selectin, ICAM-1, VCAM-1, IL-4, IL-10, IL-13, TNF-α, IFN-γ and TGF-β markers were expressed and their quantitative analysis showed significant differences compared to that of the control samples (Figure 2). Notably, endothelial markers were repeatedly observed in the endothelial cells of the lung parenchyma in patients with fatal YF (Figure 3 and Figure 4).

Among the correlations analysed, a positive correlation was observed between E-selectin and VCAM-1 (r = 0.0774, *p* = 0.010), TGF-β and E-selectin (r = 0.877, *p* =0.003), IFN-γ and P-selectin (r = 0.826, *p* = 0.005), TGF-β and VCAM-1 (r = 0.714, *p* = 0.027), TNF-α and IFN-γ (r = 0.764, *p* = 0.015), IL-13 and TNF-α (r = 0.798, *p* = 0.008), IL-4 and IL-10 (r = 0.714, *p* = 0.024), IL-4 and TGF-β (r = 0.840, *p* = 0.004) and moderate correlation was observed between P-selectin and TNF-α (r = 0.682, *p* = 0.034) (Figure 5).

## 4. Discussion

YF is an acute viral disease with great public health impact due to its ability to cause classic hemorrhagic fever. Although the immunological mechanisms related to YF progression in humans are not yet fully understood, increased vascular permeability in severe YF caused by the interaction of several factors, including the systemic and unbalanced inflammatory cytokine response and the expression of endothelial markers (E- selectin, P-selectin, intercellular adhesion molecule-1, vascular cell adhesion molecule-1), is known [1,4].

In patients with YF, pulmonary histopathology has revealed mainly alveolar hemorrhage and oedema [7,9] and similar complications have been identified in NHP with YF [12]. These classic histopathological findings in patients with YF are similar to that of our study.

In a study involving squirrel monkeys (*Saimiri* spp.), after six days of YFV infection, severe hemorrhage, oedema, congestion, and mononuclear inflammatory infiltration were observed in the lungs. Among these, hemorrhage and oedema were the main cause of death [8]. Such results corroborate our findings, indicating that in fatal cases of human yellow fever, as well as in the liver, the lung being an extremely vascularized organ, the disproportionality of the inflammatory response can result in the loss of endothelial structure that can result in a severe hemorrhage (Figure 1D).

In this context, the response induced by cytokines, such as TNF-α, IFN-γ, TGF-β, Fas ligand, CD4+ and CD8+ T cells, natural killer cells, macrophages and neutrophils, essential for triggering immunopathogenesis of liver lesions in YF [3,13].

Increased ICAM-1 and VCAM-1 expression was observed in the lung parenchyma. The increase in the number of these markers occurred mainly in the areas of inflammatory infiltration located near the vascular endothelium.

A striking feature of fatal cases of YF is the presence of an inflammatory infiltrate, in which the vascular endothelium contributes to the recruitment of defense cells [1]. This was evidenced by the analysis of endothelial marker expression in our study.

The expression of ICAM-1 and VCAM-1 in endothelial cells points to the activation of the endothelium, which is a fundamental step for leukocyte activation and endothelial inflammation during the immune response [14,15].

Here, the high levels of ICAM-1 and VCAM-1 in pulmonary tissues can be attributed to the stimulation of the immune response. Notably, an exacerbated expression of TNF-α and IFN-γ is responsible for endothelium activation and the induction of adhesion molecule expression [13,14].

Since elevated levels of E-selectin and P-selectin were noted in the lung parenchyma, the next step in our study was to understand the behaviour of these adhesion molecules.

E-selectin expression is induced by cytokines, such as TNF-α, IFN-γ and IL-1, and E-selectin presence in the endothelium may vary according to the stimulation triggered by these mediators of the inflammatory response; E-selectin expression occurs maximally after four hours of stimulation by pro-inflammatory cytokines [16,17].

E-selectin is one of the main factors responsible for the adhesion, signaling, capture and rolling of blood neutrophils that target the tissue in infectious diseases [18]. These findings strengthen our hypothesis that the presence or deficiency of adhesion molecules in the endothelium can directly influence the number of existing immune cells in the tissue and change the profile of immune response cells found at the site of inflammation.

P-selectin is present mainly in platelets and the endothelium and is translocated to the cell surface when activated by histamine, thrombin and superoxide anions [19]. An increase in P-selectin expression, when stimulated by the cytokine IL-4, has also been demonstrated [16].

E-selectin and P-selectin act similarly with regard to lymphocyte recruitment. Furthermore, both the selectins facilitate neutrophil migration towards inflamed tissues. E-selectin stabilizes the endothelial junctional integrity, which helps reduce cell passage velocity and assists in the first step of neutrophil migration. A deficit in the migration of neutrophils toward the injured tissue was observed in studies wherein P-selectin was blocked, thus corroborating the role of P-selectin in the migration of polymorphonuclear cells [20].

Our findings agree with those of other studies, in which a significant increase in the expression of endothelial adhesion molecules, E-selectin, P-selectin, ICAM-1 and VCAM-1, in patients with fatal YFV infection is described [1]. In addition, the positive association of E-selectin with IFN-γ demonstrates that the cytokine can contribute synergistically to increase the expression of the adhesion molecule in the vascular endothelium and influence the triggering of the inflammatory process in fatal cases of human yellow fever.

Here, quantitative analysis showed a significant increase in TNF-α and IFN-γ expression. This increase can be attributed to the involvement of TNF-α and IFN-γ in the cell injury and viral clearance process, potentiating inflammation and consequently manifesting the Th1 response to inhibit viral replication in the lung parenchyma of patients with fatal YF.

TNF-α plays a vital role in mediating lung injury, inducing and coordinating the activation of a network of adhesion molecules, production of inflammatory cytokines and chemokines in the lung, necessary to trigger the body’s defense mechanisms.

IFN-γ is secreted by T and NK cells during the Th1-mediated immune response. IFN-γ functions as a pro-inflammatory cytokine that directly inhibits viral replication by activating effector immune cells and increasing antigen presentation [21].

The immuno-expression of the anti-inflammatory cytokines IL-4, IL-10, and IL-13 was elevated in the lung parenchyma, suggesting their contribution to inflammation reduction modulating Th2 response and tissue repair.

IL-10 inhibits inflammatory macrophage activation, class II MHC expression and cytokine response, such as TNF-α, IFN-γ and IL-12, which is important for triggering the microbicidal response, in addition to having the ability to regulate hemodynamic parameters, leukocyte recruitment, reducing the rolling and adherence of leukocytes in the microcirculation and reducing the accumulation of leukocytes in the lung, as well as the secretion of chemotactic molecules [22,23]. Thus, the increase in IL-10 expression noted in our study may be associated with leukocyte regulation in pulmonary microcirculation.

With respect to the vascular barrier, IL-4 and IL-13 activate endothelial cells to express VCAM-1, leading to the adhesion of leukocytes expressing VLA-4 [24,25].

TGF-β, a strong inhibitor of the cellular immune response, is responsible for initiating an anti-inflammatory environment and is a strong inducer of apoptosis [13,26]. Thus, the increase in TGF-β noted in our study may be attributed to the tissue repair mechanism that responds to harmful infectious stimuli.

We performed correlation analyses to better understand the effects of cell adhesion molecules, cytokines and growth factors. Our results demonstrated a statistically significant positive correlation between E-selectin and VCAM-1 (r = 0.7744, *p* = 0.0107), which strengthens the link between the expression of these molecules and the induction of pro-inflammatory cytokines. TNF-α and IFN-γ positively regulate the expression of E-selectin and VCAM-1 [14,27].

The correlation between IL-4 and IL-10 (r = 0.7145, *p* = 0.0241) demonstrates that the relationship between the markers corroborates to a synergism of response anti-inflammatory that can negatively regulate the production of pro-inflammatory mediators, such as TNF-α, IFN-γ and IL-1 in monocytes [28,29].

Regarding the association between IL-4 and TGF-β (r = 0.8401, *p* = 0.0048), our finding reinforces that the synergistic action of IL-4 and TGF-β can influence in the repair response stimulating the production of factors that induce collagenase production and fibroblast differentiation in tissue lesions [30,31].

The moderate correlation observed for P-selectin and TNF-α (r = 0.6829, *p* = 0.0344) agrees with the findings of studies by Gotsch et al. [32] and Weller et al. [33], in which P-selectin is upregulated by TNF-α in endothelial cells.

The correlation between TNF-α and IFN-γ (r = 0.7643, *p* = 0.0151) noted in our study strengthens the synergistic link between cytokine expression and adhesion molecule induction. It is interesting to observe that, according to the histopathological alterations, the massive presence of alveolar inflammatory infiltrate in fatal cases of human yellow fever (Figure 1B) indicates that both TNF-α and IFN-γ may be influencing this process through the recruitment of leukocytes, as well as an increase in the expression of adhesion molecules [34,35].

The positive correlation observed in our study between IL-13 and TNF-α (r = 0.7982, *p* = 0.0080), TGF-β and E-selectin (r = 0.8778, *p* = 0.0036), IFN-γ and P-selectin (r = 0.8265, *p* = 0.0058), and TGF-β and VCAM-1 (r = 0.7141, *p* = 0.0274) is in contrast with the findings of previous studies. IL-13 has been demonstrated to inhibit the production of different cytokines, including TNF-α, in monocytes/macrophages [36,37]. TGF-β inhibits the expression of E-selectin in human endothelial cells and consequently decreases the adhesion of leukocytes in these cells [38,39]. IFN-γ suppresses P-selectin expression in activated human endothelial cells [40]. TGF-β has been shown to inhibit VCAM-1 expression in endothelial cells [41].

## 5. Conclusions

The findings of our study may lead to novel approaches related to cell signaling and immune responses in YFV infection. However, in-depth investigations are required to clarify the mechanisms involved and the correlation between other cytokines, endothelial adhesion molecules, and transcription factors in the immunopathogenesis of YF.

Thus, in severe hemorrhagic diseases, such as YF, an extremely damaging immune response is observed along with a modified vascular endothelium, which plays an important role in adhesion molecules, cytokine storms and the intense recruitment of defense cells triggered in the lung in response to viral aggression (Figure 6). Furthermore, our results strongly suggest that the contribution of these factors is associated with the intense and severe in situ lung damage observed in YF, which culminates in the fatal outcome of the disease.

## Figures and Tables

**Figure 1 viruses-14-02379-f001:**
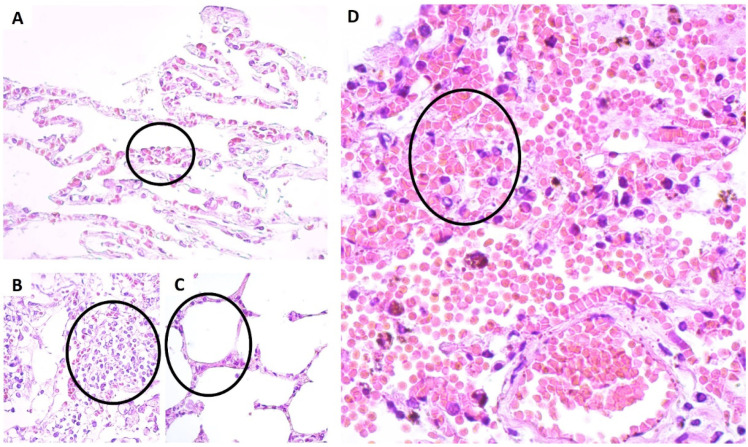
Histopathological changes in the lung tissue of patients with fatal yellow fever compared to that of control lung tissue. (**A**) Congestion (black circle); (**B**) Presence of inflammatory infiltrate (black circle); (**C**) Preservation of alveoli in the negative control (black circle); (**D**) Hemorrhage severe (black circle). Hematoxylin-Eosin. Magnification: 400×.

**Figure 2 viruses-14-02379-f002:**
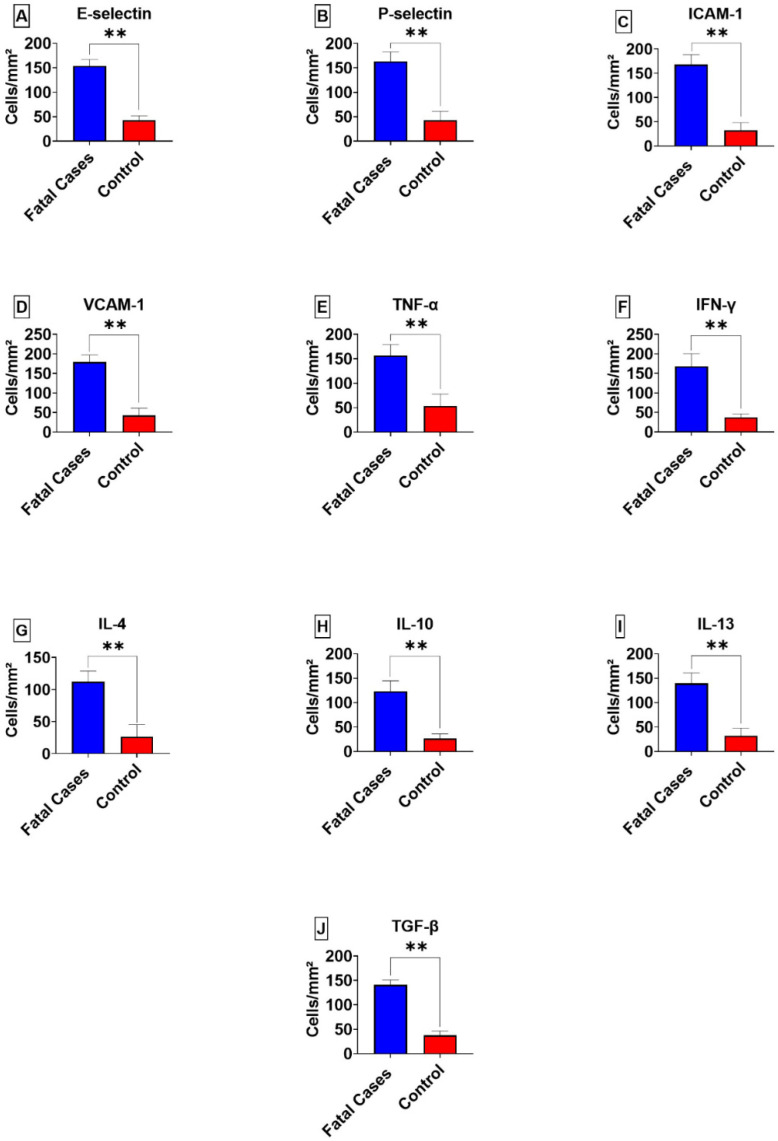
Quantitative analysis of the endothelial phenotype and expression of cytokines in the lung parenchyma of patients (10 cases) with fatal yellow fever and controls (three cases) (**A**–**J**). Mann-Whitney *p* ≤ 0.005 **.

**Figure 3 viruses-14-02379-f003:**
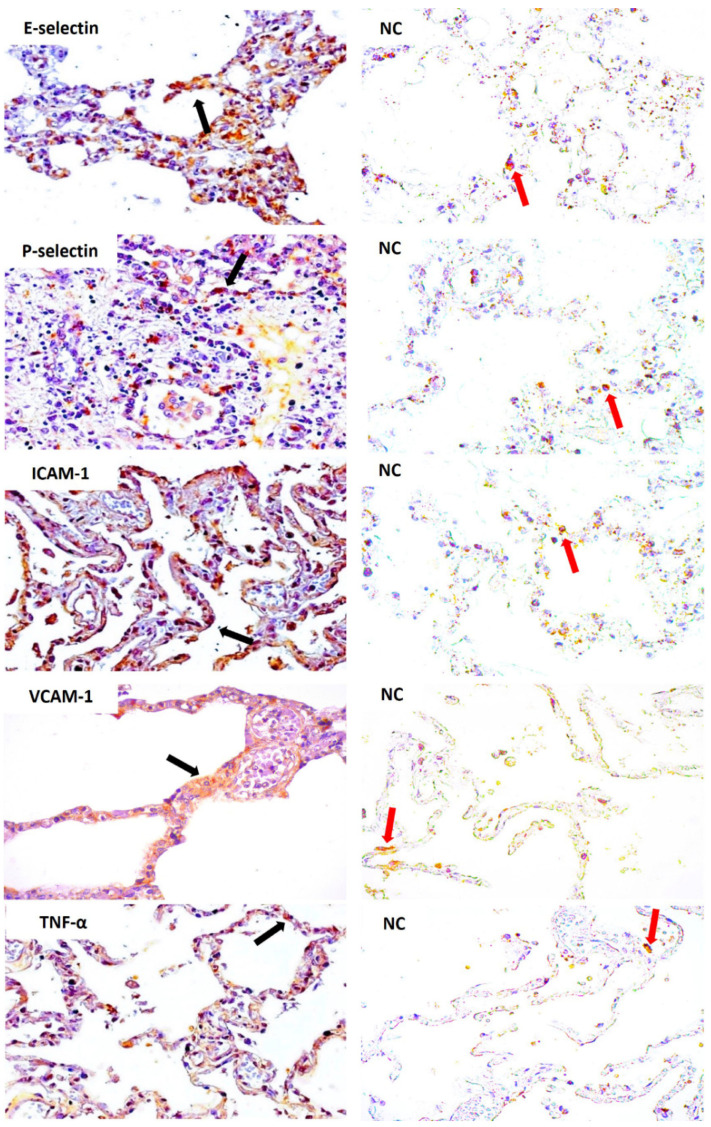
Immunostaining pattern for E-selectin, P-selectin, ICAM-1, VCAM-1 and TNF-α in the lung alveoli (black arrow) of patients with fatal yellow fever and negative controls. CN: Negative control and immunostaining in the lung alveoli (red arrow). Magnification: 400×.

**Figure 4 viruses-14-02379-f004:**
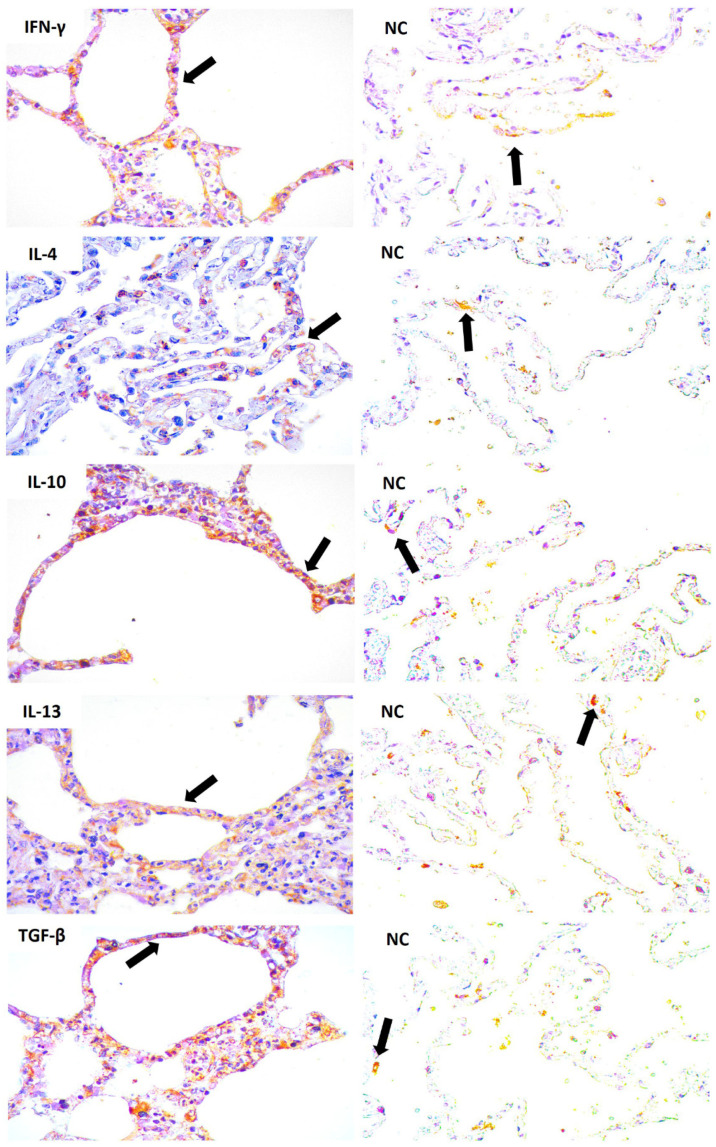
Positive immunostaining pattern for IFN-γ, IL-4, IL-10, IL-13, and TGF-β in the lung alveoli (black arrow) of patients with fatal yellow fever and negative controls. NC: Negative control and Immunostaining in the lung alveoli (black arrow). Magnification: 400×.

**Figure 5 viruses-14-02379-f005:**
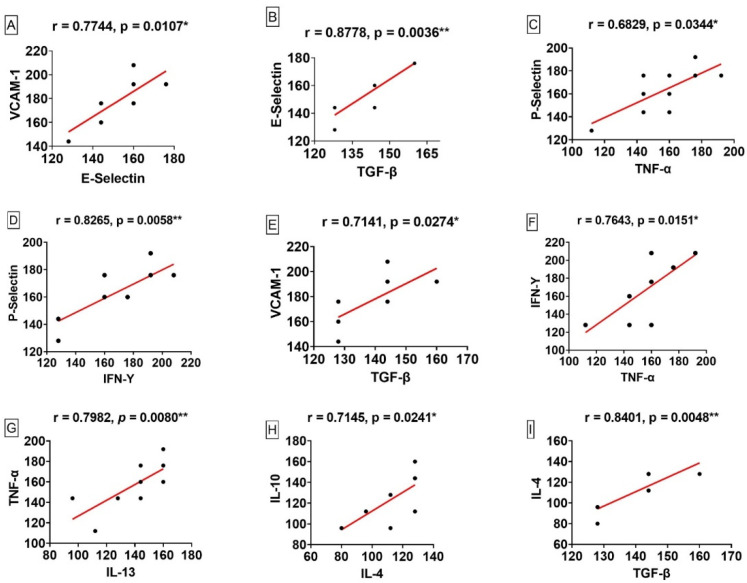
Linear correlation between markers in fatal cases of human yellow fever (**A**–**I**). * *p* ≤ 0.05, ** *p* ≤ 0.005.

**Figure 6 viruses-14-02379-f006:**
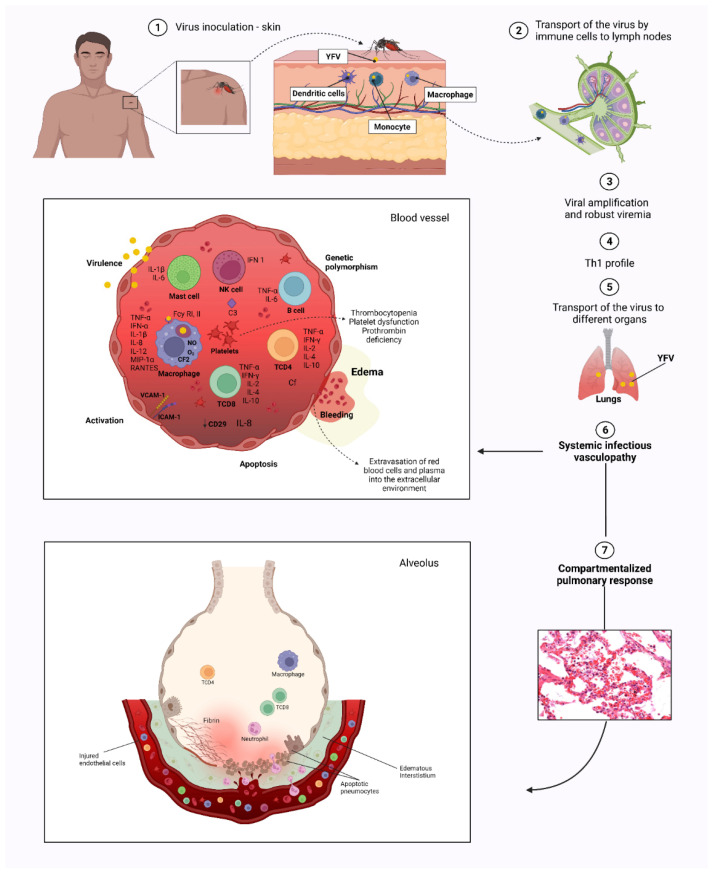
Severe hemorrhagic in YF and damaging immune response with a modified vascular endothelium, which plays an important role in adhesion molecules, cytokine storms, and the intense recruitment of defense cells triggered in the lung in response to viral aggression.

**Table 1 viruses-14-02379-t001:** Characterization of yellow fever patients according to their precedence, age and gender, illness time (I.T.).

Case	State	Sex	Age	I.T. (Days)	Patient	Year
1	TO	M	30	N.I.	494/00	2000
2	GO	M	23	N.I.	074/07	2007
3	GO	F	63	2	043/08	2008
4	DF	M	55	-	088/08	2008
5	GO	M	42	N.I.	095/08	2008
6	DF	M	35	N.I.	154/08	2008
7	GO	M	35	N.I.	062/16	2016
8	PB	M	-	N.I.	102/16	2016
9	GO	M	15	7	346/16	2016
10	GO	M	27	1	369/16	2016

N.I.—not included. (-)—not found, M—Male; F—Female; GO—Goiás; PB—Paraíba; TO—Tocantins; DF—Distrito Federal.

## Data Availability

The data used to support the findings of this study are included within the article.

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
