# Peer review of "New Insights into the Mechanism of Immune-Mediated Tissue Injury in Yellow Fever: The Role of Immunopathological and Endothelial Alterations in the Human Lung Parenchyma"

_viruses, 2022, doi:10.3390/v14112379_

Round 1

Reviewer 1 Report

In the paper, `New Insights into The Mechanism of Immune-Mediated Tis- 2 sue Injury In Yellow Fever: The Role of Immunopathological  and Endothelial Alterations in the Human Lung Parenchyma`. In their analysis, they use ihc to quantify the level of immune response markers in lung tissues of YFV victims. In general, the paper has interesting findings, but lacks novelty as most of these markers are likely related to cytokine storms, a well-known cause of mortality of YFV. Similar studies have already been conducted in different tissues, finding very similar results , as an example of https://doi.org/10.1016/j.virusres.2005.08.019 , https://www.ncbi.nlm.nih.gov/pmc/articles/PMC8779659/ . Overall, the paper is of some interest, but lacks true novelty as these results are more or less expected for the know outcomes of the disease

The writing could be much improved to avoid redundancy and correct misspelling

Minor
abstract line 22 `by immunohistochemistry method, 22 based on complex formation biotin streptavidin peroxidase` - rewrite

Abstract line 23 - `There was a difference in the expression of markers between fatal cases of YF and control samples` - unnecessary

General comment for abstract. Should carry more information about the results and conclusions, and less

Figure 2 and Table 2 are the same data

Figure 2 would be much more interesting if sample dispersion is also visualizable

Author Response

Revisor 1

Open Review

English language and style

( ) Extensive editing of English language and style required
(x) Moderate English changes required
( ) English language and style are fine/minor spell check required
( ) I don't feel qualified to judge about the English language and style

Yes

Can be improved

Must be improved

Not applicable

Does the introduction provide sufficient background and include all relevant references?

( )

(x)

( )

( )

Are all the cited references relevant to the research?

( )

(x)

( )

( )

Is the research design appropriate?

(x)

( )

( )

( )

Are the methods adequately described?

(x)

( )

( )

( )

Are the results clearly presented?

( )

(x)

( )

( )

Are the conclusions supported by the results?

( )

(x)

( )

( )

Comments and Suggestions for Authors

In the paper, `New Insights into The Mechanism of Immune-Mediated Tis- 2 sue Injury In Yellow Fever: The Role of Immunopathological  and Endothelial Alterations in the Human Lung Parenchyma`. In their analysis, they use ihc to quantify the level of immune response markers in lung tissues of YFV victims. In general, the paper has interesting findings, but lacks novelty as most of these markers are likely related to cytokine storms, a well-known cause of mortality of YFV. Similar studies have already been conducted in different tissues, finding very similar results , as an example of https://doi.org/10.1016/j.virusres.2005.08.019 , https://www.ncbi.nlm.nih.gov/pmc/articles/PMC8779659/ . Overall, the paper is of some interest, but lacks true novelty as these results are more or less expected for the know outcomes of the disease

R: Thank very much by comments. However, we believe that this manuscript advances in the study of the pathogen-host relationship, especially with regard to the compartmentalized immune response in shock organs, such as in the case of the lung, which has an extremely vascularized functional circulation and that in this case this is the first study that proposes to characterize an in situ immune response pattern that is directly correlated with histopathological changes and with the fatal outcome in YFV-positive cases. Although we agree that there are other studies and different bodies, the investigation of the tissue immune response (where the agent interacts with the host's immune system) is important to explain the pathogenesis of the disease. This relationship is not the same when comparing different organs because each organ has a characteristic immune tissue organization, which prevents the findings from being compared between different organs. Please see: PMID: 33174908, PMID: 16872652, PMID: 35704977, PMID: 16278000.

The writing could be much improved to avoid redundancy and correct misspelling

R: Thank very much by comments. The paper was corrected as requested.

Minor
abstract line 22 `by immunohistochemistry method, 22 based on complex formation biotin streptavidin peroxidase` - rewrite

R: Thank very much by comments. The paper was corrected as requested.

Abstract line 23 - `There was a difference in the expression of markers between fatal cases of YF and control samples` - unnecessary

R: Thank very much by comments. The paper was corrected as requested.

General comment for abstract. Should carry more information about the results and conclusions, and less

R: Thank very much by comments. The paper was corrected as requested.

Figure 2 and Table 2 are the same data

R: Thank very much by comments.Table 2 was removed as suggested.

Figure 2 would be much more interesting if sample dispersion is also visualizable

R: Thank very much by comments. We kindly request that the reviewer reconsider the presentation of the graph, maintaining the current presentation as it facilitates the visualization of the reader who is not an expert in the area.

Reviewer 2 Report

In this manuscript the authors (Danielle Barbosa Vasconcelos et al.) describe their results of their histopathological and immunohistochemical investigation of lung tissue from ten individuals succumbing to yellow fever virus (YFV) infection. All patients apparently had hemorrhagic pneumonia, although it is hard to ascertain based on the few microphotos how acute/subacute the hemorrhage was, as it is not possible to discern erythrophagocytosis by alveolar macrophages or hemosiderin-accumulation in these cells and interstitial monocyte/macrophages as would be expected if the diapedesis or frank hemorrhage was of several days’ development. Length of clinical illness is only provided for three of the cases and no other clinical information is provided. Contrary to what the authors claim infiltration of inflammatory leukocytes seems modest at best (compared to what respiratory viruses incites it would be classified as very mild), and neutrophils are not discernable at the micrograph resolution provided.

The description of the immunohistochemical is lacking details such as washing steps, the buffer used for dilution of antibodies etc. Even more importantly, there is no mention of positive and negative controls for the various antibodies. This is of concerns as there would appear to be some non-specific reaction in many of the panels in Figures 3 and 4 – mainly we transudates in the alveoli, but also reaction with what might be interpreted as alveolar macrophages, suggesting that Fc-receptors have not been appropriately blocked. It is also disconcerting that no signal is seen in the “negative control”, yet in Figure 2 and Table 2 there are cell counts for the control for all markers. At least some should be visible in the control panels even if the numbers are lower. It would also be appropriate to show higher magnifications of the positive cells, so the reader can be confident about the cell type expressing the various molecules.

It is also intriguing that the supposedly positive signal is red to orange to occasionally reddish brown when DAB was used as a substrate. In the hands of this reviewer, DAB should result in a brown to very dark-brown signal – and certainly not red.

In order to draw conclusions about the pathogenesis (if even possible at all), it would be necessary to have virological data for these patients too: (i) were they viremic? (ii) was virus isolated from any tissues, including the lung, or tested for by RT-PCR ? (iii) has an attempt been made to detect viral antigen in the lungs by immunohistochemistry? The latter would certainly strengthen the interpretation of the immunological markers.

While the authors provide information about the microscope used in section 2.4, it is unclear whether they used image analysis software or manually counted the cells. If the former, details should be provided. If the latter, then make that clear.

It is questionable whether all the permutations of correlations between the various target molecules are relevant. As a matter of fact, and as also pointed out by the authors in the Discussion, they likely are not, since some of the associations make no biological sense at all. Moreover, without virological data (virus antigen detection or virus isolation or viral RNA detection), it all becomes utterly hypothetical/speculative. In other words, the results appear to be over-interpreted and statements like “This corroborates our findings of quantifying this crucial factor………..” (lines 222-3) or “suggesting their contribution to inflammatory reduction” (line 228), or “demonstrates that the relationship between the markers corroborates the increase in the suppressor response ………” (lines 252-255) are inappropriate.

While it might be correct, in some settings, that TNF-alpha plays a role in lung injury and induce apoptosis, then it is all context dependent. The authors should consider to do immunohistochemistry for activated caspase 3 to demonstrate apoptosis. It is not discernible in the photos provided, but that is not to say it is not present – but it would require visualization to make the claim of occurrence.

And in the paragraph lines 256-260, the authors first state that IL-4 and TGF-beta can interfere with repair processes and then go on to say that these two cytokines can stimulate production of factors that promote healing/repair, makes it all very confusing and utterly inconclusive.

The manuscript could be further improved by addressing the following:

·       The abstract goes into details of cytokine background etc, which is inappropriate – it should focus on aim, approach, results and conclusions.

·       Line 32: an inflammatory process is not “constructed” but rather elicited.

·       Lines 43-44: the first sentence in the paragraph should either be rephrased or deleted all together.

·       Line 71-73: something seems to be missing, as the authors jumps straight from stating something about respiratory viruses to have a parenthesis with ages.

·       Line 82: change to “For the detection of phenotypic markers and cytokines, antibodies specific for E-selectin ……….”

·       Line 89 and elsewhere: change ‘immunostaining’ to “immunolabeling”

·       Line 97: change to “Antibody binding was visualized using a chromogen solution………”

·       Line 127: replace ‘recurrently’ with “repeatedly”

·       Figure 2 and Table 2 presumably present the same data. One presentation of the data should suffice, so it is suggested to either delete the table or place it in a supplementary file.

·       Lines 172-3: what do the authors mean by “the virus and the inflammatory factors modify its endothelial structure……”. The sentence needs rephrasing, but at the same time points to the lack of virology data: how can the authors claim that the virus has influence on endothelial structure, when they have not presented evidence of YFV presence?

·       Line 191: no, E-selectin expression is not ‘mediated’ by cytokines, but might be “induced” by such molecules.

·       Line 284: it seems rather exaggerated to talk about “an extremely damaging immune response”, as that is not what the authors have demonstrated. Nor is there the “intense recruitment of defence cells” (lines 285-6) – at least that is not apparent in the microphotos provided. 

E

Author Response

Revisor 2

Open Review

English language and style

(x) Extensive editing of English language and style required
( ) Moderate English changes required
( ) English language and style are fine/minor spell check required
( ) I don't feel qualified to judge about the English language and style

Yes

Can be improved

Must be improved

Not applicable

Does the introduction provide sufficient background and include all relevant references?

( )

(x)

( )

( )

Are all the cited references relevant to the research?

( )

(x)

( )

( )

Is the research design appropriate?

( )

( )

(x)

( )

Are the methods adequately described?

( )

( )

(x)

( )

Are the results clearly presented?

( )

( )

(x)

( )

Are the conclusions supported by the results?

( )

( )

(x)

( )

Comments and Suggestions for Authors

In this manuscript the authors (Danielle Barbosa Vasconcelos et al.) describe their results of their histopathological and immunohistochemical investigation of lung tissue from ten individuals succumbing to yellow fever virus (YFV) infection. All patients apparently had hemorrhagic pneumonia, although it is hard to ascertain based on the few microphotos how acute/subacute the hemorrhage was, as it is not possible to discern erythrophagocytosis by alveolar macrophages or hemosiderin-accumulation in these cells and interstitial monocyte/macrophages as would be expected if the diapedesis or frank hemorrhage was of several days’ development. Length of clinical illness is only provided for three of the cases and no other clinical information is provided. Contrary to what the authors claim infiltration of inflammatory leukocytes seems modest at best (compared to what respiratory viruses incites it would be classified as very mild), and neutrophils are not discernable at the micrograph resolution provided.

R: Thank very much by comments. We modified the images in Figure 1 and included the histopathological alterations well evidenced in the fatal cases as suggested.

The description of the immunohistochemical is lacking details such as washing steps, the buffer used for dilution of antibodies etc. Even more importantly, there is no mention of positive and negative controls for the various antibodies. This is of concerns as there would appear to be some non-specific reaction in many of the panels in Figures 3 and 4 – mainly we transudates in the alveoli, but also reaction with what might be interpreted as alveolar macrophages, suggesting that Fc-receptors have not been appropriately blocked. It is also disconcerting that no signal is seen in the “negative control”, yet in Figure 2 and Table 2 there are cell counts for the control for all markers. At least some should be visible in the control panels even if the numbers are lower. It would also be appropriate to show higher magnifications of the positive cells, so the reader can be confident about the cell type expressing the various molecules.

R: Thank very much by comments. We modified figures 3 and 4 and included images of the control with labeling as suggested and also included information on the dilution of antibodies as suggested.

It is also intriguing that the supposedly positive signal is red to orange to occasionally reddish brown when DAB was used as a substrate. In the hands of this reviewer, DAB should result in a brown to very dark-brown signal – and certainly not red.

R: Thank very much by comments. We emphasize that, in this case, the marking visualization tone is related according to the camera filter and the program used. In this case, we emphasize that because it is a hot and cold filter, the color of the marking tends to vary from brown to reddish orange. In addition, we reinforce the fact that for the fatal cases positive for YFV and for the negative control, we insert arrows in the images to highlight the specific and very punctual marking of cells that may be expressing cytokines or adhesion molecules for better identification. of reviewers and readers.

In order to draw conclusions about the pathogenesis (if even possible at all), it would be necessary to have virological data for these patients too: (i) were they viremic? (ii) was virus isolated from any tissues, including the lung, or tested for by RT-PCR ? (iii) has an attempt been made to detect viral antigen in the lungs by immunohistochemistry? The latter would certainly strengthen the interpretation of the immunological markers.

R: Thank very much by comments. The qPCR was performed in several organs including liver, spleen, blood for diagnosis and case definition. We know that viremia, in the advanced phase of the disease, is not marked and the pathophysiological process stems from the host's immune response, which is the purpose of our work. Additional information can be found in the manuscript on the diagnosis.

While the authors provide information about the microscope used in section 2.4, it is unclear whether they used image analysis software or manually counted the cells. If the former, details should be provided. If the latter, then make that clear.

R: Thank very much by comments. The count was performed manually from the selection of 10 fields in the lung parenchyma as described in the methodology. Additional information about the microscope and the software has been included in the text.

It is questionable whether all the permutations of correlations between the various target molecules are relevant. As a matter of fact, and as also pointed out by the authors in the Discussion, they likely are not, since some of the associations make no biological sense at all. Moreover, without virological data (virus antigen detection or virus isolation or viral RNA detection), it all becomes utterly hypothetical/speculative. In other words, the results appear to be over-interpreted and statements like “This corroborates our findings of quantifying this crucial factor………..” (lines 222-3) or “suggesting their contribution to inflammatory reduction” (line 228), or “demonstrates that the relationship between the markers corroborates the increase in the suppressor response ………” (lines 252-255) are inappropriate.

R: Thank very much by comments. We modified the text as suggested.

While it might be correct, in some settings, that TNF-alpha plays a role in lung injury and induce apoptosis, then it is all context dependent. The authors should consider to do immunohistochemistry for activated caspase 3 to demonstrate apoptosis. It is not discernible in the photos provided, but that is not to say it is not present – but it would require visualization to make the claim of occurrence.

R: Thank very much by comments. We modified the text as suggested.

And in the paragraph lines 256-260, the authors first state that IL-4 and TGF-beta can interfere with repair processes and then go on to say that these two cytokines can stimulate production of factors that promote healing/repair, makes it all very confusing and utterly inconclusive.

R: Thank very much by comments. We modified the text as suggested.

The manuscript could be further improved by addressing the following:

  • The abstract goes into details of cytokine background etc, which is inappropriate – it should focus on aim, approach, results and conclusions.

R: Thank very much by comments. We modified the text as suggested.

  • Line 32: an inflammatory process is not “constructed” but rather elicited.

R: Thank very much by comments. We modified the text as suggested.

  • Lines 43-44: the first sentence in the paragraph should either be rephrased or deleted all together.

R: Thank very much by comments. We modified the text as suggested.

  • Line 71-73: something seems to be missing, as the authors jumps straight from stating something about respiratory viruses to have a parenthesis with ages.

R: Thank very much by comments. We modified the text as suggested.

  • Line 82: change to “For the detection of phenotypic markers and cytokines, antibodies specific for E-selectin ……….”

R: Thank very much by comments. We modified the text as suggested.

  • Line 89 and elsewhere: change ‘immunostaining’ to “immunolabeling”

R: Thank very much by comments. We modified the text as suggested.

  • Line 97: change to “Antibody binding was visualized using a chromogen solution………”

R: Thank very much by comments. We modified the text as suggested.

  • Line 127: replace ‘recurrently’ with “repeatedly”

R: Thank very much by comments. We modified the text as suggested.

  • Figure 2 and Table 2 presumably present the same data. One presentation of the data should suffice, so it is suggested to either delete the table or place it in a supplementary file.

R: We appreciate the suggestion and removed table 2 as suggested.

  • Lines 172-3: what do the authors mean by “the virus and the inflammatory factors modify its endothelial structure……”. The sentence needs rephrasing, but at the same time points to the lack of virology data: how can the authors claim that the virus has influence on endothelial structure, when they have not presented evidence of YFV presence?

R: Thank very much by comments. We modified the text as suggested.

  • Line 191: no, E-selectin expression is not ‘mediated’ by cytokines, but might be “induced” by such molecules.

R: Thank very much by comments. We modified the text as suggested.

  • Line 284: it seems rather exaggerated to talk about “an extremely damaging immune response”, as that is not what the authors have demonstrated. Nor is there the “intense recruitment of defence cells” (lines 285-6) – at least that is not apparent in the microphotos provided. 

R: Thank very much by comments. We modified the text as suggested.

Reviewer 3 Report

In this study, the researchers perform immunohistochemical stainings to compare lung parenchyma tissue of patients that died due to yellow fever disease with control samples. They find that all 10 evaluated immunological markers are elevated compared to the control situation, which contributes to the observed pathological condition by activating the endothelium and increasing inflammation.

The study shines a very interesting light on the less studied effects on the induced inflammatory pathways of YFV in lung tissue. However, the manuscript could be better structured, more details on data representation should be included and additional control results could be revealed as well.

40 It should be mentioned that severe YF disease only occurs in a relatively small number of cases and the majority of infections is characterized by milder symptoms

46 are these cited data from blood or tissue samples?

51 strange sentence

69 Is it possible to correlate lung tissue cytokine levels with viral copy number?

81 From the way this method section is written now it is unclear if these stainings were performed on separate samples of each individual patient or simultaneously for all ten markers. In case of the first way, how were the samples appointed to a specific antibody? How many sample slides were analysed for each patient in total?

110 for which part of the analysis were medians calculated?

114 histhopathological findings: are these results based solely on visual microscopical evaluation, did the researchers perform the H&E stainings? How is the distinction between lymphocytes, plasma cells and neutrophils made?

124 Could the researchers include a control marker that does not change in YFV compared to control samples?

Figure 1 Are these representative results of one patient or more? Could this be included in the figure caption?

Figure 2 and Table 1: represented in the figure and the table are redundant. Perhaps only the figure should be included, if desired the table can be included in supplementary information but this adds little to the data. The number of samples should be included in the figure caption.

Figures 3 and 4: these figures are somewhat unclear to the reader. What exactly are the arrows indicating? Could more arrows be included to clarify? Are these representative images of one patient for each marker or not? What is indicated in the NC? How is the expression of the markers in patients and NC quantified using these images, could this perhaps be included in a schematic figure?

Figure 5 Why are not all 10 data points included for each marker? What about the other correlations, if they are not significant they could be included in the supplementary information? The type of analysis should also be included in the figure caption.

General remark on the discussion:

Could this be better structured in paragraphs to improve the reading flow? First the observed results are stated and linked to 1) the pathological symptoms and 2) existing literature. Then the correlations are discussed. In a last part the aberrant observations are observed. Could the authors elaborate somewhat on the possible explanation?

Typos

23 of biotin

153 and 261: 0.6829

248 facilitate

291 hemorrhagic disease

Author Response

Revisor 3

Open Review

English language and style

( ) Extensive editing of English language and style required
( ) Moderate English changes required
(x) English language and style are fine/minor spell check required
( ) I don't feel qualified to judge about the English language and style

Yes

Can be improved

Must be improved

Not applicable

Does the introduction provide sufficient background and include all relevant references?

(x)

( )

( )

( )

Are all the cited references relevant to the research?

(x)

( )

( )

( )

Is the research design appropriate?

( )

(x)

( )

( )

Are the methods adequately described?

( )

(x)

( )

( )

Are the results clearly presented?

( )

(x)

( )

( )

Are the conclusions supported by the results?

(x)

( )

( )

( )

Comments and Suggestions for Authors

In this study, the researchers perform immunohistochemical stainings to compare lung parenchyma tissue of patients that died due to yellow fever disease with control samples. They find that all 10 evaluated immunological markers are elevated compared to the control situation, which contributes to the observed pathological condition by activating the endothelium and increasing inflammation.

R: Thank very much by comments.

The study shines a very interesting light on the less studied effects on the induced inflammatory pathways of YFV in lung tissue. However, the manuscript could be better structured, more details on data representation should be included and additional control results could be revealed as well.

R: Thank very much by comments.

40 It should be mentioned that severe YF disease only occurs in a relatively small number of cases and the majority of infections is characterized by milder symptoms

R: Thank very much by comments. We modified the text as suggested.

46 are these cited data from blood or tissue samples?

R: Thank very much by comments. We modified the text as suggested.

51 strange sentence

R: Thank very much by comments. We modified the text as suggested.

69 Is it possible to correlate lung tissue cytokine levels with viral copy number?

R: Thank very much by comments. As we used samples from fatal cases and stored in 10% formalin and embedded in paraffin, this correlation was not possible.

81 From the way this method section is written now it is unclear if these stainings were performed on separate samples of each individual patient or simultaneously for all ten markers. In case of the first way, how were the samples appointed to a specific antibody? How many sample slides were analysed for each patient in total?

R: Thank very much by comments. For each sample or patient, immunostaining for specific antibodies was performed on different slides according to the defined panel. One slide for each marker as described in other works in the literature, please see: PMID: 33174908, PMID: 16872652, PMID: 35704977, PMID: 16278000.

110 for which part of the analysis were medians calculated?

R: Thank very much by comments. We removed this analysis from the methodology.

114 histhopathological findings: are these results based solely on visual microscopical evaluation, did the researchers perform the H&E stainings? How is the distinction between lymphocytes, plasma cells and neutrophils made?

R: Thank very much by comments. We performed HE stains and modified figure 1.

124 Could the researchers include a control marker that does not change in YFV compared to control samples?

R: Thank very much by comments. Negative controls were included for the analysis of our results for the main arboviruses circulating in Brazil. The figures have been modified to make the result clearer.

Figure 1 Are these representative results of one patient or more? Could this be included in the figure caption?

R: Thank very much by comments. The figures are representative of the main findings that were common in the 10 fatal cases.

Figure 2 and Table 1: represented in the figure and the table are redundant. Perhaps only the figure should be included, if desired the table can be included in supplementary information but this adds little to the data. The number of samples should be included in the figure caption.

R: Thank very much by comments. We have included additional information in the figure legend to detail the importance of the information contained in each one.

Figures 3 and 4: these figures are somewhat unclear to the reader. What exactly are the arrows indicating? Could more arrows be included to clarify? Are these representative images of one patient for each marker or not? What is indicated in the NC? How is the expression of the markers in patients and NC quantified using these images, could this perhaps be included in a schematic figure?

R: Thank very much by comments. We modified the figure, added the arrows and additional information.

Figure 5 Why are not all 10 data points included for each marker? What about the other correlations, if they are not significant they could be included in the supplementary information? The type of analysis should also be included in the figure caption.

R: Thank very much by comments. We included in our results the correlations that were most representative and that revealed statistical significance. For the others that are not presented in the manuscript, the value of r was very small and without statistical significance.

General remark on the discussion:

Could this be better structured in paragraphs to improve the reading flow? First the observed results are stated and linked to 1) the pathological symptoms and 2) existing literature. Then the correlations are discussed. In a last part the aberrant observations are observed. Could the authors elaborate somewhat on the possible explanation?

R: Thank you for your comments, we have made some changes to the discussion text to make the manuscript clearer.

Typos

23 of biotin

R: Thank very much by comments. We rewrite the sentence as suggested.

153 and 261: 0.6829

R: Thank very much by comments. Corrections were made as suggested.

248 facilitate

291 hemorrhagic disease

R: Thank very much by comments. Corrections were made as suggested.

Round 2

Reviewer 1 Report

The authors addressed most of concerns properly

Author Response

The authors addressed most of concerns properly

R: Thank very much by comments.

Reviewer 2 Report

While some modifications have been made to the manuscript, there are still no information on virus (RNA, infectious virus or viral antigen) in the lung samples and therefore the whole discussion of the role of the various phenotypic markers and cytokine expression becomes pointless.

It remains impossible to judge the extent of hemorrhage and inflammation in the lung parenchyma, let alone assess the type of leukocyte infiltration, based on the few microphotos provided. What proportion of the lungs were affected by hemorrhage and inflammation? How representative of the whole lung are the microphotos shown in Figure 1?

The IHC figures (Figures 3 and 4) remain unconvincing. Only one panel is changed in Figure 3 (VCAM-1) and it is no more convincing than the original microphoto. Viewing the figures at 100% on the computer screen, it is mostly impossible to discern what the arrows are pointing to and enlargement (on the computer screen), does not aid in the identification of cell types. There remains a lot of background signaling, i.e., signal-to-noise ratio is just too high to make these microphotos convincing.

The authors are still overinterpreting the results and exaggerating the degree in inflammation. A better executed study is required to address the aspects of the pathogenesis of YFV infection that the authors claim to do in this manuscript.

Author Response

Comments and Suggestions for Authors

While some modifications have been made to the manuscript, there are still no information on virus (RNA, infectious virus or viral antigen) in the lung samples and therefore the whole discussion of the role of the various phenotypic markers and cytokine expression becomes pointless.

R: Thank very much by comments. We believe that the pansystemic effect that is triggered is crucial to modulate the in situ immune response and lead to tissue damage by the direct action of the virus and especially, in the most advanced stages of the disease, the immune-mediated response. The lesions found in other organs such as kidney and liver also depend on the exacerbated immune response against the virus that leads to vascular damage and tissue damage. Few studies published in the literature have already demonstrated the relationship between the presence of the viral antigen and lung injury (please see PMID: 35322047, PMID: 30802414, PMID: 31087672). We believe that the results found in our study are of great value and we reinforce that several techniques served as support for the confirmation of the disease as described in the methodology.

It remains impossible to judge the extent of hemorrhage and inflammation in the lung parenchyma, let alone assess the type of leukocyte infiltration, based on the few microphotos provided. What proportion of the lungs were affected by hemorrhage and inflammation? How representative of the whole lung are the microphotos shown in Figure 1?

R: Thank very much by comments. We selected the images that are most representative and that summarize the histopathological findings of our series and that corroborate the description of other studies carried out with humans (please see: PMID: 30802414; PMID: 31329590).

The IHC figures (Figures 3 and 4) remain unconvincing. Only one panel is changed in Figure 3 (VCAM-1) and it is no more convincing than the original microphoto. Viewing the figures at 100% on the computer screen, it is mostly impossible to discern what the arrows are pointing to and enlargement (on the computer screen), does not aid in the identification of cell types. There remains a lot of background signaling, i.e., signal-to-noise ratio is just too high to make these microphotos convincing.

R: Thank very much by comments. The images are in high resolution and, for the photomicrographs, we were careful to select images illustrative of the very punctual marking in the alveoli. We emphasize that the figures are in the pixel standard requested by the Viruses.

The authors are still overinterpreting the results and exaggerating the degree in inflammation. A better executed study is required to address the aspects of the pathogenesis of YFV infection that the authors claim to do in this manuscript.

R: Thank very much by comments. We understand and appreciate your inquiries. However, we believe that the study and manuscript as it stands advances the discussion regarding YFV infection and the relationship between compartmentalized in situ immune response and endothelial damage in the lung and lung parenchyma.

Reviewer 3 Report

The authors have addressed some of the points made by this reviewer. However, there still are some points to be cleared out.

Line 79: I understand that correlation between viral load and histopathological findings is not possible, but here it is stated that diagnosis was confirmed with RT-qPCR. Could this be quantitatively be included in Table 1?

Figures 3 and 4; it looks like the authors have just inserted other microscopy pictures compared to the first revision, in stead of adding arrows to make the picture more clear?

Figure 5: the authors still do not give an explanation for the correlations where <10 cases are shown.

246: Figure 1D

Author Response

Comments and Suggestions for Authors

The authors have addressed some of the points made by this reviewer. However, there still are some points to be cleared out.

R: Thank very much by comments.

Line 79: I understand that correlation between viral load and histopathological findings is not possible, but here it is stated that diagnosis was confirmed with RT-qPCR. Could this be quantitatively be included in Table 1?

R: Thank very much by comments. In the text found in the methodology, we emphasize that the confirmation of the diagnosis was supported by other additional techniques and that both in the serum and in the viscera, in the case of PCR, we did not measure the viral load, because in this case our interest was only to know whether the sample was positive for the virus or not. Several studies have already highlighted the tropism of both the vaccine antigen and the virus for lung tissue, however this relationship, in advanced stages of the disease (which constitutes our sample), in the toxemic phase, there is a decrease in viral load in several organs and induction of injury eminently immune-mediated (please see PMID: 35322047, PMID: 30802414, PMID: 31087672, PMID: 35056050, PMID: 35746675, PMID: 35805137, PMID: 16278000).

Figures 3 and 4; it looks like the authors have just inserted other microscopy pictures compared to the first revision, in stead of adding arrows to make the picture more clear?

R: Thank very much by comments. For the new figures, we included arrows that indicate very punctual marking and, as requested by other reviewers, the changes made in the control group point to arrows that also show very punctual marking in alveolar cells. This information is included in the legend of the figures to facilitate the reader's understanding.

Figure 5: the authors still do not give an explanation for the correlations where <10 cases are shown.

R: Thank very much by comments. For figure 5, the 10 cases were used. However, we emphasize that for all graphs (A-I), the points that represent an ordered pair (x,y) of a case and do not appear, have the same mean value of another case.

246: Figure 1D

R: Thank very much by comments. We have included additional information on the subject in the manuscript.
